# Multiple Technologies Combined to Analyze the Changes of Odor and Taste in Daokou Braised Chicken during Processing

**DOI:** 10.3390/foods11070963

**Published:** 2022-03-26

**Authors:** Feili Zhan, Lingxia Sun, Gaiming Zhao, Miaoyun Li, Chaozhi Zhu

**Affiliations:** 1College of Food Science and Technology, Henan Agricultural University, Zhengzhou 450002, China; yrd2018@stu.henau.edu.cn (F.Z.); gmzhao@126.com (G.Z.); limy7476@126.com (M.L.); zhuchaozhi66@163.com (C.Z.); 2Key Laboratory of Meat Processing and Quality and Safety Control of Henan Province, Zhengzhou 450002, China

**Keywords:** Daokou braised chicken, odor, taste, GC-MS, GC-IMS, e-nose, e-tongue

## Abstract

This study analyzed the changes of odor and taste in Daokou braised chicken during processing by GC-MS, GC-IMS, e-nose and e-tongue. The 75 and 55 volatile compounds identified in Daokou braised chicken by GC-MS and GC-IMS, respectively, included hydrocarbons, aldehydes, alcohols, terpenes, ketones, heterocyclics, esters, acids and phenols; among them, aldehydes, alcohols and ketones were the most abundant. The number and proportion of volatile compounds in Daokou braised chicken changed significantly (*p* < 0.05) in the process. The proportion of volatile compounds with animal fatty odor, such as aldehydes and alcohols, decreased, while that of esters, ketones and terpenes from spices with fruity fragrance increased, especially in the braising stage. An e-nose showed that the odor intensities of sulfur-containing and nitrogen oxide compounds were higher (*p* < 0.05) after the braising stage, but weakened after 2 h braising. An e-tongue showed that saltiness and richness increased significantly (*p* < 0.05) after braising. The results of these four techniques showed that braising promoted the release of flavor compounds, and was beneficial to salt penetration and umami release. However, long braising could lead to weakened flavor intensity and the introduction of bitterness and astringency. This study also found that GC-IMS and e-nose were more sensitive to trace compounds such as sulfur-containing and nitrogen oxide compounds, esters, acids and phenolics in Daokou braised chicken than GC-MS. The use of multiple technologies could provide more comprehensive flavor profiles for Daokou braised chicken during processing. This study provides insights into the control of flavor of Daokou braised chicken, and may be of practical relevance for the poultry industry.

## 1. Introduction

Daokou braised chicken is a typical representative of Chinese traditional sauce and brine meat products. It was first created over 300 years ago, during the Qing Dynasty (1661) [1]. It is famous for its distinctive flavor, beautiful shape, bright color and tender, boneless meat. It is also known as Beijing Braised Duck and Jinhua Ham [2], and is deeply loved by consumers. The processing of Daokou braised chicken mainly includes modeling, coloring, frying and stewing. Eight kinds of spices are added in the process and supplemented with aged soup, resulting in products with delicious taste and rich aroma.

Flavor is composed of odor and taste originating from both volatile and nonvolatile compounds, and plays a critical role in food quality. In recent years, some studies have shown that the volatile flavor components of braised chicken are mainly aldehydes, alcohols and terpenes, etc. [3]. Other compounds that contribute significantly to the formation of taste are mainly nonvolatile substance such as free amino acids, small peptides, nucleotides, etc. [4]. Yao et al. [5] analyzed the flavor formation of Dezhou braised chicken during production and found that 2-ethylhexanol was the key flavor chemical of raw chicken. Yan Duan et al. [6] found the main odor-active constituents of the Dezhou braised chicken were carbonyl compounds, while Liu et al. [7] found that IMP (inosinic acid), Glu (glutamic acid), Lys (L-lysine), and sodium chloride were the main compounds contributing to taste. Intelligent sensory techniques such as electronic nose (e-nose) and electronic tongue (e-tongue) can quickly analyze and recognize the odor and taste characteristics of foods. The e-nose can identify flavors quickly and nondestructively by analyzing flavor profilse [8]. The e-tongue can be used for rapid, accurate, qualitative and quantitative analyses of flavors [9]. Yuan et al. [10] analyzed the diferences of aroma and taste of Tan sheep using an e-nose and e-tongue. Chromatographic techniques such as gas chromatography-mass spectrometry (GC-MS) and gas chromatography-ion mobility spectrometry (GC-IMS) are currently important research tools for analyzing food flavor [11,12]. GC-MS is a classical technique for the detection of organic compounds, which could provide detailed information on volatile compounds owing to its standards-based reference database. GC-IMS is a powerful technique in meat flavor analyses, with the characteristics of fast response, high sensitivity, convenient operation, high flux and test results visualization in flavor research [13]. GC-IMS does not need high temperature treatments for sample determinations, which is advantageous, as these can reduce the quality of flavor information [14]. These techniques have the advantages of high sensitivity, good selectivity and objective accuracy, and can be used in combination to obtain more comprehensive, reliable and scientific information on food flavor. Duan et al. [15] found the flavor differences of two salmon species by GC-IMS combined with an electronic nose and electronic tongue. Di Wang et al. [16] investigated the flavor characteristics of wind duck at different aging times using an e-nose, e-tongue and GC-IMS. Chen et al. [17] studied the effect of thermal processing methods on the aroma profiles of acidity regulator-treated tilapia muscles using an e-nose, HS-SPME-GC-MS, and HS-GC-IMS, obtaining useful knowledges regarding the flavors of aquatic products and meats. Tian et al. [18] studied the effect of salt content on the flavor of dry-cured pork using an e-tongue and GC-IMS, and determined the appropriate salt content. Guo et al. [19] established the characteristic volatile fingerprints of fresh and dried matsutake by HS-GC-IMS and HS-SPME-GC-MS, and found significant differences in terms of flavor.

This study investigated changes of odor and taste by using intelligent sensory techniques combined with GC-MS and GC-IMS chromatography, with the aim of clarifying the flavor evolution during the processing of Daokou braised chicken, so as to provide a reference for flavor quality control during production. From a practical point of view, a combination of factors has a positive impact on the flavor quality of Daokou braised chicken; knowledge thereof has the potential to attract more consumers and increase economic benefits.

## 2. Materials and Methods

The raw materials selected were chilled San Huang chicken and rooster, each weighing about 1–1.5 kg. Chicken, salt, spices, honey and other materials were purchased from Shuang Hui meat store, Zheng Zhou city, Henan province, China. And alkane standards (C6–C26) were obtained from Sigma Company (St. Louis, MO, USA). The samples required for this experiment were prepared according to the traditional processing of Daokou braised chicken. First, the raw chicken was washed and shaped. Second, honey water (the mass ratio of honey to water was 1:3) was evenly spread on the chicken and dried naturally, before frying (155 °C, approx. 2 min) in oil (soybean oil) until the skin of the chicken turned golden brown. Third, eight spices (tangerine peel, angelica dahurica, tsaoko, cardamom, amomum, galangal, clove, cinnamon) were weighed according to the traditional recipe, wrapped in gauze and then put, along with salt, raw chicken and water, successively into the pot (ATUF/DR, Beijing, China Abbott) and stewed. At this stage, the water was first boiled and then cooled to 95 °C, for 2 h of braising. Samples of chicken meat were collected for analysis after the four key stages (Figure 1): raw chicken, frying, braised for 1 h and 2 h, named L1, L2, L3, L4 respectively. Selected skinless chicken meat from the breast and thigh, obtained from three randomly selected chickens at each of the four stages, were chopped and then individually mixed with the samples from each stage. The samples were stored at −40 °C and equilibrated to room temperature before extraction.

### 2.1. GC-MS Analysis

First, a 5 g thawed sample was placed in a 20 mL headspace bottle. Extraction of the flavor in the sample was then carried out by using solid phase microextraction (SPME) fiber (50/30 μm CAR/PDMS/DVB; Merck company, Kenilworth, NJ, USA). GC-MS analysis of the extracts was performed on a SCION SQ 456-GC (Bruker, Birrika, MA, USA) equipped with a DB-Wax column (30 m × 0.25 mm i.d., film thickness 0.25 μm,). The method of extraction of flavor compounds and GC-MS analysis is described in the literature [20]. The retention time of each compound was converted to a retention index (RI) using n-alkanes (C6–C26) as references. Volatile compounds were identified based on comparisons of their mass spectra with those in the NIST08 database and by matching the RI values with those reported in the literature. The peak area normalization method was used for quantitative analyses and the relative contents of volatile substances were obtained.

### 2.2. GC-IMS Analysis 

Analyses of flavor compounds were carried out on a GC-IMS Flavor Spec^®^ (G.A.S, Beijing, China). First, a 4 g sample was placed in a 20 mL headspace injection vial, incubated at 60 °C for 10 min and then injected into the sample for analysis. The temperature of head space injection needle was 70 °C and the injection volume was 300 μL. GC was equipped with SE-54 capillary column (15 m × 0.53 mm); the column temperature was 40 °C and carrier gas/drift gas was N2. The flow rate of the whole phase was as follows: 2 mL/min for 2 min, 100 mL/min in 8 min, 150 mL/min in 5 min and 150 mL/min for 15 min. The analysis time was 30 min. The analytes were eluted and driven into an IMS ionization chamber for ionization by a radioactive ionization source (3H) of (300 MBq activity) in positive ion mode. Then, the ions were placed into a drift tube (5 cm in length) through a shutter grid which operated at a constant voltage of 400 V/cm, a temperature of 45 °C and a drift gas (nitrogen gas) flow rate of 150 mL/min. 

### 2.3. E-Nose Analysis

E-nose analysis was performed using a PEN3 E-nose (Win Muster Air Sense Analytics Inc., Schwerin, Germany). This e-nose has 10 sensors. The response characteristics of each sensor are as follows: W1C (aromatic), W5S (oxynitride), W3C (aromatic), W6S (hydrogen), W5C (alenes and aromatic compounds), W1S (alkanes), W1W (hydrogen sulfide), W2S (alcohols and partial aromatic compounds), W2W (aromatic compounds and organic sulfides) and W3S (alkane) [21]. First, a 10 g sample was placed into a 150 mL triangle bottle, sealed and heated in water bath at 35 °C for 5 min. A probe was inserted into the sealed bottle to exude the flavor through a drainage membrane. The test conditions were as follows: sample test time was 80 s: cleaning time was 120 s; the internal flow rate was 300 mL/min; and the sample flow rate was 300 mL/min.

### 2.4. E-Tongue Analysis

Taste analyses were performed using a SA402BE-tongue (Insent Inc., Atsugi-shi, Japan). First, 50 g of minced meat with 200 mL deionized water were placed into a 250 mL conical flask and mixed well. This was then placed in a water bath at 50 °C for 30 min. The mixed solution was then centrifuged at 3000 r/min (Rotational Speed) for 10 min to obtain the supernatant. Then, the supernatant was filtered through qualitative filter paper, and the solution to be tested was obtained. The e-tongue sensor system was equipped with seven basic taste indexes, i.e., umami, astringency, saltiness, bitterness, richness, aftertaste-astringency, and aftertaste-bitterness. Sensors were washed in a cleaning solution for 90 s and reference solutions (120 s + 120 s) to make sure the sensors had been thoroughly cleaned. The sensors were balanced for 30 s to ensure the baselines were standardized before taking measurements. The duration of each measurement was 30 s.

### 2.5. Data Treatment

Analysis of variance (ANOVA) was used to evaluate differences among samples using SPSS software (Version 26, IBM Inc., Stamford, CA, USA); *p* < 0.05 indicated statistically significant differences using Duncan’s multiple range tests. A clustering heat map of the result of GC-MS was plotted using the HIPLOP (Bioinformatics opensource community) software. The data of GC-IMS were analyzed by LAV software (version 2.2.1), GC × IMS Library Search, and some plugins. LAV and GC-IMS Library Search were respectively used for the quantitative and qualitative analyses of substances. The Reporter plugin, Gallery plot plugin and dynamic PCA plugin were respectively used to determine spectral differences, for fingerprint comparisons and dynamic principal component analyses. The score figures of the principal components analysis (PCA) using the e-nose and e-tongue were both analyzed using Origin (Version 2022, Origin Lab Corporation, Northampton, MA, USA).

## 3. Results and Discussion

### 3.1. Analysis of GC-MS

#### 3.1.1. Volatile Compounds in Different Processing Stages 

Volatile compounds in Daokou braised chicken at different processing stages were detected by GC-MS, as shown in Table 1. The relative contents of alcohols changed significantly during processing; these relative contents were second only to aldehydes (*p* < 0.05). In particular, the content of enols was the highest in frying stage, but this significantly decreased after 2 h of braising, which may have been due to conversion into aldehydes, ketones and other substances [22]. It has been reported that the oxidation of polyunsaturated fatty acids leads to the production of alcohols during heating processes. Saturated alcohol has little effect on the flavor of the braised chicken because of its high odor threshold (500–20,000 μg/kg), while unsaturated alcohol has a certain effect on flavor because of its low odor threshold. Additionally, 1-Octene-3-ol and (E)-2-Octenol contributed significantly to the flavor of the Daokou braised chicken due to their high concentrations and low odor thresholds. The former compound could confer a mushroom aroma upon Daokou braised chicken [23].

Ketones are generally associated with creamy and fruity flavor characteristics [17], which have little contribution to meat flavor owing to their high odor threshold. Ketones may arise from the oxidation of fat. The relative content of ketones increased with processing, and increased significantly in the stewing stage (*p* < 0.05) This indicates that the thermal processes had a positive effect on the formation of ketone compounds. The relative contents of 2,3-Octanedione and 3-Hydroxy-2-butanone were higher than those of other ketones, which may contribute to harmonizing the flavor of Daokou braised chicken.

The proportion of aldehydes was higher in each stage of processing, especially nonanal, hexanal, octanal and heptanal, which played important roles in the characteristic flavor because of their low odor thresholds [24]. Although the content of enal and dienal were lower, they were characteristic flavor compounds owing to their extremely low thresholds when chicken fat was heated [25]. The Strecker degradation of leucine may promote the production of 3-Methydil-butanal [26]. Additionally, 3-Methyl-butanal was detected in Dezhou braised chicken [5,6]. Previous research indicated that aldehydes are mainly derived from lipid oxidation. Aldehydes changed significantly during the processing (*p* < 0.05). The content of aldehydes was the highest in raw meat, lowest after braising for 1 h and then increased significantly after braising for 2 h. Heptanal, octanal and nonanal decreased significantly (*p* < 0.05) during the frying stage. Aldehydes mostly have a fatty odor, and their proportion directly affected the odor of the Daokou braised chicken. The main volatile compounds of raw chicken were aldehydes with animal fat odor if no heat processing was applied. Increasingly volatile components were produced under the heat of frying and braising, while the proportion of aldehydes decreased after frying and braising for 1 h. Similarly, aldehydes continued to be produced with lipid oxidation and the degradation of certain amino acids, the proportion of which increased (while remaining significantly lower than in raw chicken), giving Daokou braised chicken a unique and harmonious odor. 

Hydrocarbons have a high odor threshold and are mostly odorless or with a faint odor [27]; as such, they have little direct effect upon the flavor of braised chicken. Aliphatic hydrocarbons may be produced by the oxidative degradation of lipids, and aromatic hydrocarbons may be produced by the oxidation of free amino acids with aromatic groups. However, the relative content of hydrocarbons in the samples changed significantly during processing (*p* < 0.05), and increased significantly during frying and braising, from 0.55% of raw chicken to 8.47% after braising for 1 h, indicating that hot processing is beneficial to the formation of hydrocarbons in braised chicken. Octahydro-4,7-methano-1H-indene, 3-Ethyl-2-methyl-1,3-hexadiene and (E)-2,2-dimethyl-4-decene were detected in the samples; these compounds are precursors of ketones and aldehydes, and likely have effects on flavor.

All terpenes except 1,8-Cineole were generated from spices added during the braising process, indicating that the braising stage plays a crucial role in the formation of terpenes [8]. However, the relative contents of terpenes significantly decreased after 2 h of braising (*p* < 0.05), compared with 1 h. Linalool and Trans-p-mentha-1(7),8-Dien-2-ol were only detected after braising for 2 h, indicating that braising for a long time is beneficial to the release of flavor compounds from spices [5]. Terpenes mostly provide herbal and floral fruity [28] flavors, which contribute greatly to the characteristic flavor of Daokou braised chicken due to their low odor thresholds. Additionally, 1,8-Cineole provides camphor and cool herbal notes, Limonene a pleasant lemon aroma and β-Fenchol a wild floral aroma of cloves and vanilla. The relative contents of 1,8-Cineole, Limonene and β-Fenchol were higher, indicating that they may be important contributors to the delicate flavor of Daokou braised chicken.

Two heterocyclic compounds both 2-Pentyl-furan and 5-Ethenyltetrahydro-2-furanmethanol were detected in the braised chicken: 2-Pentyl furan is the main flavor compound for thermally processed food, derived from linoleic acid and other n-6 fatty acids, with relatively low threshold and plant aroma [28,29]. The content of this compound increases (*p* < 0.05) throughout processing. Only one ester, hexyl acetate, was detected in the frying and braising stage, and its concentration did not vary (*p* > 0.05). With the exceptions of lactone [30] and thioester, esters have high thresholds and low abundances in meat, contributing little to the flavor of meat products. Oxime-, methoxy-phenyl- and N,N-Dibutylformamide were also detected. Oxime-, methoxy-phenyl- existed throughout processing, and may have originated from the material itself [31] or from contaminants in the environment or veterinary drugs. N,N-Dibutylformamide may have originated from pesticide residue.

Volatile compounds in Daokou braised chicken at different processing stages were detected by GC-MS. As shown in Figure 1, 40, 49, 57 and 63 volatile compounds were respectively detected in samples L1, L2, L3, L4. The types of volatile compounds in Daokou braised chicken increased significantly with the progress of thermal processing. A total of 75 volatile compounds (R. match and F. match were both greater than 800) were detected including 20 hydrocarbons, 16 aldehydes, 15 alcohols, 10 terpenes, 9 ketones, 4 heterocyclics and 1 ester. The quantity and proportion of volatile compounds in Daokou braised chicken during processing are shown in Figure 2. The proportion of aldehydes and alcohols remained high throughout processing. With continued processing, the number of volatile components increased significantly. Compared with L1, the number of volatile compounds increased dramatically in the other samples. Heat treatments such as frying, braising, etc. are major contributors to the formation of flavor substances in meat products [32,33].

#### 3.1.2. Analysis of Heat Map of Volatile Compounds

In order to further study the change of odor and its component in the processing of Daokou braised chicken, the relative percentage contents of 75 volatile compounds in Table 1 were standardized as variables, and then heat map was drawn and hierarchical clustering analysis was used to cluster volatile compounds (Figure 3). As shown in Figure 3, the volatile compounds could be divided into six categories. The darker the red, the higher the content, and the darker the blue, the lower the content. Class I compounds mainly appeared in raw chicken, followed by the frying stage, including aldehydes and alcohols, and mainly providing fatty odor. Class II compounds mainly appeared in the frying stage, followed by braising for 1 h, including alcohols, aldehydes, ketones and one heterocyclic compound, providing fatty and fruity fragrance. Almost all compounds in class III mainly appeared in braising for 2 h, including terpenes, aromatics, enaldehydes, branched-chain aldehydes and enols, probably the characteristic odor compounds of Daokou braised chicken, which provided fatty, fruity, herbal and woody fragrance [28], etc. And most of these compounds with low odor threshold, contributed the braised chicken strong and rich aroma characteristics. There were 8 volatile compounds in class IV, which appeared in the frying and braising stages, especially the frying stage, mainly included alkanes and alcohols. Compounds in class V mainly appeared in the braising stage, especially when braised for 2 h, including terpenes, ketones, alcohols and heterocyclics, which mainly provided fruity, woody and nutty fragrance [29]. Class VI compounds including hydrocarbons, terpenes and alcohol compounds, mainly appeared in braising for 1 h, which mainly provided fruity, herbal and spicy fragrance. The research results showed that stewing for 2 h will promote the formation of the characteristic flavor of Daokou braised chicken.

The odor compounds of Daokou braised chicken in different processing stages changed constantly, and the types and aroma attributes of volatile compounds more abundant, thus forming the unique odor. The volatile compounds in raw chicken mainly were from class I and presented fatty odor. In the frying stage, the volatile compounds were mainly from class II and class IV, which contributed with fatty and fruity odor. After braised for 1 h, mainly volatile compounds were from class VI, and some from class V and IV, which presented fatty, fruity, herbal and spicy fragrance. Except class I, the volatile compounds in the chicken braised for 2 h were mainly from class III and V and some from class II, IV and VI. When braising for 2 h, the compounds were more diverse, so that the braised chicken had rich and harmonious odor. The above showed that the braising process was the key processing stage for the formation of characteristic odor of Daokou braised chicken. Although the content of some compounds decreased after braising for 2 h, the braising process accelerated the release of volatile compounds, and appropriate extension of braising time was conducive to the proper proportion of volatile compounds, so as to form rich and harmonious odor. The results of four techniques showed that braising promoted the release of flavor compounds, and was the key stage for the formation of characteristic flavor in Daokou braised chicken. Although some volatile compounds decreased or disappeared after braising for 2 h, the types and characteristics of flavor compounds in chicken were more diverse, giving it a rich and harmonious aroma.

### 3.2. Analysis of GC-IMS

#### 3.2.1. Topographic Analysis 

As shown in Figure 4A,B the topographic plots obtained for the four samples analyzed, which overlay the 2D spectra. It was a two-dimensional map in which Y axis represented the retention time in the chromatographic column (in seconds), and X axis represented the drift time in the drift tube (in milliseconds). Each spot in the topographic plot represented a chemical compound. The color intensity signified the concentration of compound, which meant lower content by white and higher content by red [16]. The deeper the color, the higher concentration it had. As could be seen from the overall qualitative map (Figure 4A), a total of 55 volatile substances were identified in samples from the four processing stages, including monomer and dimer. With the depth of processing, the number and color of spots in the samples changed significantly, the number increased and the color was more intense (Figure 4B), indicating that both number and content of volatile compounds had increased. The spot distribution of braised chicken was higher than that of raw and fried chicken, indicating that the types of volatile compounds in the braising stage were abundant, which was consistent with the results of GC-MS analysis in 3.1 of this paper and Yao et al. [5]. However, the spot distribution was similar for 1 h and 2 h of braising, indicating that the release of volatile compounds gradually reached equilibrium after braising for 1 h.

#### 3.2.2. Fingerprints Analysis

For a clearer comparison of the volatile compounds during the different processing stages, signals of the volatile compounds were selected to develop a comprehensive spectral fingerprint using a gallery plot plug-in (Qian et al., 2021) (Figure 5). In Figure 5, rows represent the signal peaks from the samples and columns represent the signal peak of the same volatile compounds in different samples. As shown in Figure 5, obvious differences among the samples were clearly observed. The parts with obvious differences in fingerprints were detected and marked as “a”, “b”, “c” and “d” respectively. These distinct regions indicated that the flavor of the samples had markedly changed during processing. From Table 2 we could see the names of compounds in regions a, b, c and d.

The contents of substances in area “a” were lowest in raw chicken, gradually increasing after frying and the stewing stage, and highest in the stewing stage. These mainly included ethyl-3-methylbutanoate, methyl-2-methylbutanoate, ethyl butanoate, ethyl pentanoate, Z-3-hexenol, butanoic acid, etc. The content of substances in area “b” were detected in the highest contents in raw chicken, i.e., mainly 2-methylpropionic acid, hydroxyacetone, pentan-2 3-dione, 2-pentanone, isoamyl alcohol, 2-methylpyrazine, hexanal, P-xylene, etc. The peroxidation of fatty acid may promote the production of hexanal. These substances disappeared after thermal processing and could be regarded as characteristic volatile compounds in raw chickens. Substances in region “c” were more abundant in raw chicken and fried chicken samples but lower after the stewing stage, such as 1-butanol, 3-methyl-3-butennol, 2-hexenol, 2-methyl pyrazine, propyl acetate, benzaldehyde, pentanol, heptanal, etc. The volatile compounds in part “d” mainly appeared during braising, especially after braising for 1 h, including ethyl-3-methylbutanoate, phenylacetaldehyde, 2-penthyl-furan, ethyl propionate, A-pinene, 2-methoxyphenol, acetophenone, isovaleric acid, 2-octanone,2,3-butanediol,2-hexanone,3-methylpentanoic acid, nonanal, 2-phenylethanol, phenol-4-methyl, 1-octanol, 1,2-furanyl-ethanone, and 2-nonanone, while the contents of these compounds in raw chicken and fried chicken were extremely low. It could be clearly seen that volatile compounds were more abundant in the braising stage, with increases in the quantities of esters, ketones, 2-pentyl furan and A-pinene. It is known that 2-Pentyl furan contributes greatly to the characteristic flavor of Daokou braised chicken and that its formation is related to Maillard reactions.

### 3.3. Comparison of GC-MS and GC-IMS

Both GC-MS and GC-IMS were able to distinguish the flavor of samples at different processing stages, although GC-IMS showed the difference of flavor between samples more intuitively. As shown in Figure 6, the qualitative results of volatile compounds revealed certain differences between the two technologies. Seventy-five volatile compounds were identified by GC-MS, while 79 were detected by GC-IMS, even though only 55 were ultimately identified. Compared with GC-MS, the number of volatile compounds identified by GC-IMS was lower, which was consistent with previous reports. This might have been related to the limitation of IMS database and the inability to characterize volatile compounds [34,35]. The numbers of hydrocarbons, aldehydes and terpenes identified by GC-MS were much higher than those identified by GC-IMS, while the number of esters identified by GC-IMS was much higher than that identified by GC-IMS. Additionally, five acids and three phenolic compounds were identified by GC-IMS which were not identified by GC-MS. The contents of acids and phenols in chicken were very low. GC-MS was not sensitive to low level volatile compounds, which were difficult to detect, while GC-IMS was highly sensitive to trace volatile compounds [17,36]. Esters identified by GC-IMS were mostly with branched chains, while acids were low-grade fatty acids and fatty acids with branched chains. These compounds had relatively low boiling points, and GC-IMS was more sensitive to high-volatile compounds than GC-MS [37,38]. Note that dimers of five compounds were identified by GC-IMS, depending on the analyte concentration, their proton affinity or drift tube temperature [12,39]. Therefore, the combined use of GC-MS with GC-IMS made up for their limitations and provided more comprehensive odor profiles of the samples. This study also found that GC-MS was more sensitive to hydrocarbons, aldehydes and terpenes, while GC-IMS was more sensitive to trace compounds such as esters, acids and phenolics. The combination of GC-MS and GC-IMS was shown to be able to provide a comprehensive flavor profile for Daokou braised chicken during processing.

### 3.4. Analysis of E-Nose

PCA is a very powerful multivariate statistics method used to analyze the inherent structure of data and then express those data in such a way as to highlight their similarities and differences. In Figure 7A, the samples from different process stages are presented in PCA spatial distribution map. The contribution rates of PC1 and PC2 were 66.4% and 18.5% respectively, representing nearly 85% of the total variance. The smaller the distance between samples, the closer their quality characteristics [28]; the distance between the four samples was relatively large, indicating that the chicken odor at different stages of preparation was significantly different. The distribution of the samples in the first principal component was from L1 to L4, which showed some regular changes. The differences of samples L1 and L2 in the first principal component were obvious, as were those of samples L3 and L4 in the second principal component. The PCA loading plot of the e-nose is shown in Figure 7C. The first component (PC1) comprised 66.4%, which was positively related to the W1W, W5S, W2W, W1S, W2S, W3S and W6S sensors, and negatively related to the W1C, W3C and W5C sensors. The W5S (oxynitride) and W1W (hydrogen sulfide) sensors contributed more to sample L4, which shows that volatile compounds such as hydrocarbons and oxides were more abundant in L4 than in other samples. This result was consistent with the detection result of GC-MS, which shows that the various volatile compounds in Daokou braised chicken were more abundant after stewing.

A radar graph (Figure 7B) further confirmed the conclusion from the PCA analysis. The W5S and W1W sensors had stronger responses to the odor of samples, and the odor differences in the samples were mainly detected by sensors W5S (sensitive to nitrogen oxides compound) andW1W (sensitive to organic sulfides compounds). Although few of these compounds were identified by GC-MS and GC-IMS, they were easily detected by the e-nose sensors due to their low thresholds [40]. The response values of samples L3 and L4 at sensors W5S and W1W were higher than those of the other two samples, while there was no significant difference between samples 1 and 2, indicating that the samples in the braising stage might have had higher abundances of nitrogen oxides and organic sulfides. Note that the response value of sample L4 was lower than that of sample L3, i.e., the odor intensity after 2 h of braising was weaker than that after 1 h. These results were consistent with those of GC-MS and GC-IMS. 

### 3.5. E-Tongue Analysis

E-tongues turn electrical signals into relish signals to reflect the taste information of food. They have a small threshold of sensation, and the subjectivity of sensory evaluation can be excluded [41]. As shown in Figure 8A, the cumulative contribution rates of PC1 (74.9%) and PC2 (19.6%) were 94.5%, indicating that most taste information was reflected. Samples L1 and L2 were close to each other, indicating similar taste characteristics while samples L3 and L4 were far from each other, indicating different taste characteristics. Both samples L3 and L4 were far from samples L1 and L2, indicating that the chicken taste had changed significantly during braising.

A radar graph was also used to illustrate the changes of various tastes during processing (Figure 8B). There were significant differences (*p* < 0.05) among the samples in terms of saltiness, astringency and richness. However, no significant distinction (*p* > 0.05) was observed for bitterness, umami, bitter aftertaste and astringency aftertaste. Compared to samples L1 and L2, the saltiness, astringency and richness in samples L3 and L4 were higher (especially saltiness), and increased during the braising process. From Figure 8C, we can see that the richness and saltiness sensors contributed more to samples L3 and L4. Long braising time was beneficial to salt penetration and umami release, but could lead to bitterness and astringency. The value of astringency in this study was less than minus 2, i.e., lower than that of the reference solution, and as such, had little effect on the taste. However, we still need to pay attention to the possible adverse effects of braising for a long time on taste, and apply appropriate braising times in the preparation of Daokou braised chicken.

## 4. Conclusions

In this study, changes of odor and taste in Daokou braised chicken during processing were comprehensively analyzed by GC-MS, GC-IMS, e-nose and e-tongue. The study showed that braising promotes the release of flavor compounds, and was key for the formation of the characteristic flavor of Daokou braised chicken. Of the 75 and 55 volatile compounds identified in Daokou braised chicken by GC-MS and GC-IMS, respectively, the majority were hydrocarbons, aldehydes, alcohols, terpenes, ketones, heterocyclics, esters, acids and phenols. Aldehydes, alcohols and ketones were detected in higher proportions throughout processing, but their changes were not consistent. With the development of processing, the proportion of volatile compounds with animal fatty odor, such as aldehydes and alcohols, decreased, while the proportion of esters, ketones and terpenes from spices with fruity fragrances increased, especially in the braising stage. The results from the e-nose and e-tongue also showed that the chicken flavor changed significantly during the braising process. The e-nose results showed that oxides were more abundant after stewing, which was consistent with the detection result of GC-MS. It should be noted that braising was found to be beneficial to flavor formation, salt penetration and umami release, but that long braising times could lead to reduced flavor intensity and some bitterness. Therefore, more in-depth research on the changes of flavor compounds in braising and the critical braising time will be the main focus of future research.

## Figures and Tables

**Figure 1 foods-11-00963-f001:**
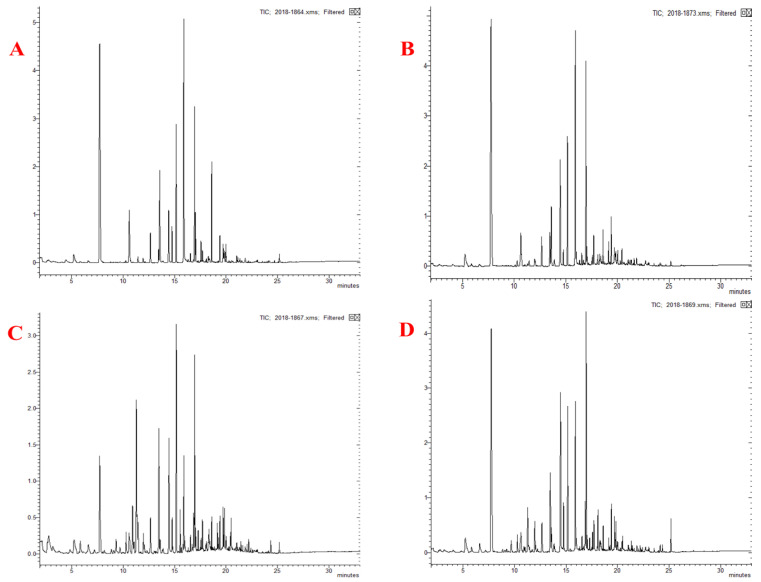
Total ion chromatograms (TICs) of volatile products in Daokou braised chicken during processing (Note: (**A**–**D**) refer to samples L1, L2, L3, L4 respectively).

**Figure 2 foods-11-00963-f002:**
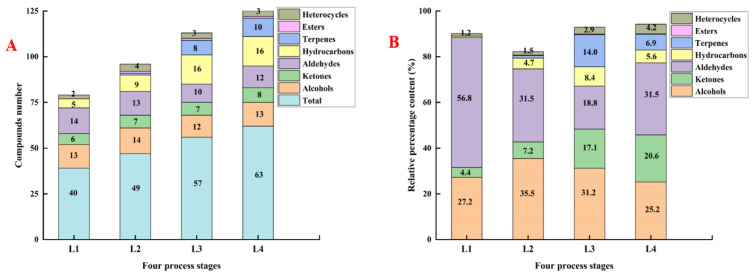
The contents and number of volatile flavor components in Daokou braised chicken during processing. (Note: (**A**,**B**) refer to the contents and number of volatile flavor components respectively).

**Figure 3 foods-11-00963-f003:**
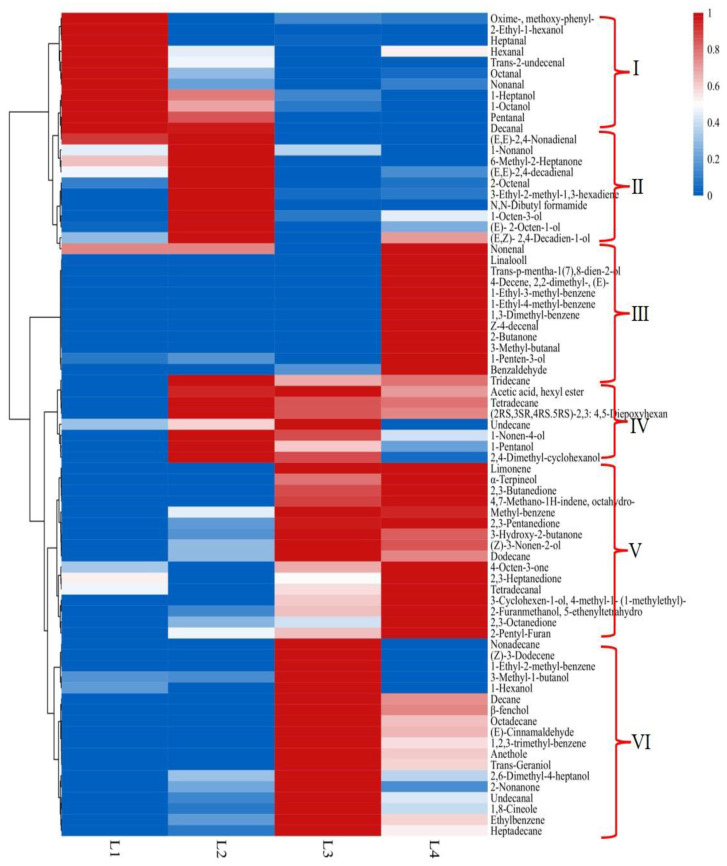
Analysis of heat map of volatile compounds during different processing stages.

**Figure 4 foods-11-00963-f004:**
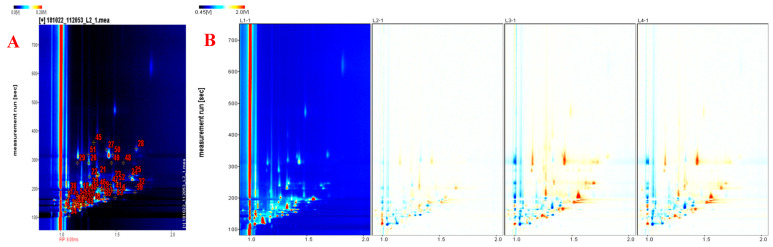
GC-IMS two-dimensional qualitative topographic plot (**A**) and topographic plot of the four processing stages (**B**).

**Figure 5 foods-11-00963-f005:**
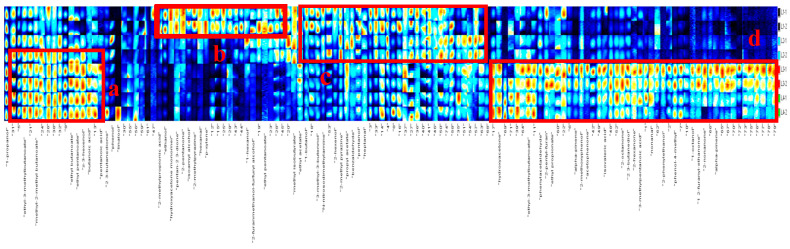
Fingerprint spectra (C) of volatile compounds in Daokou braised chicken at different processing stages. (Note: The numbers represent compounds which could not be accurately identified).

**Figure 6 foods-11-00963-f006:**
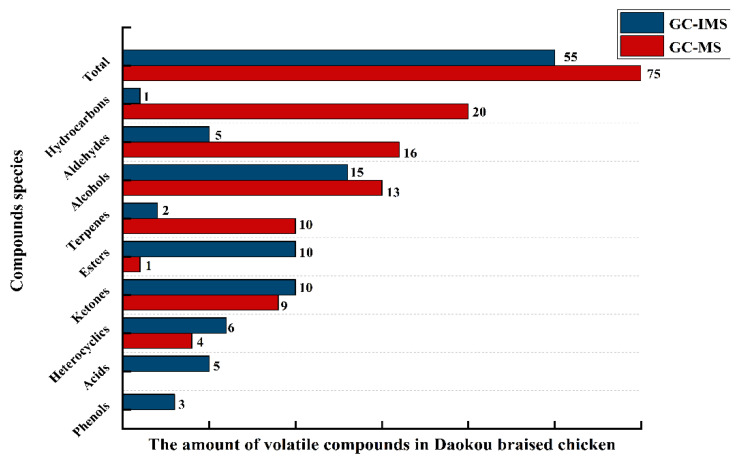
Comparison of the number of volatile flavor compounds detected by GC-MS and GC-IMS.

**Figure 7 foods-11-00963-f007:**
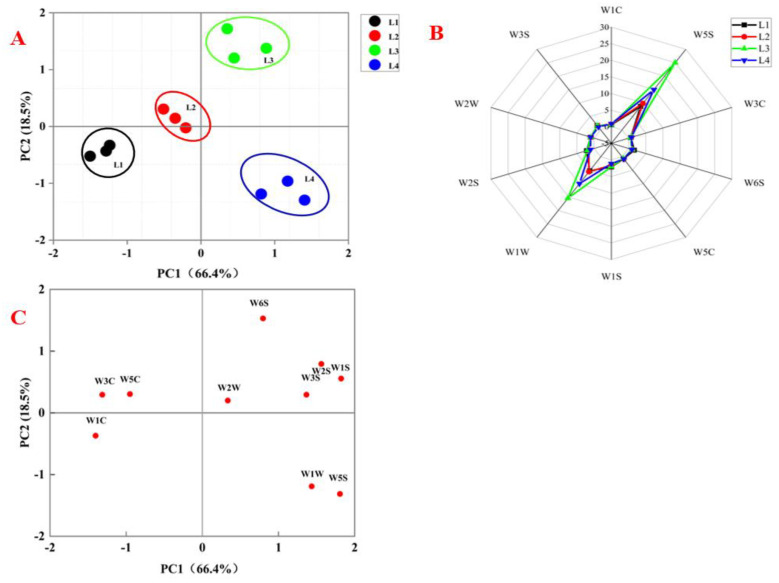
E-nose analysis. (**A**) PCA plot of the samples at different processing stages. (**B**) Radar graph of the samples at different processing stages. (**C**) PCA loading plot of the samples at different processing stages.

**Figure 8 foods-11-00963-f008:**
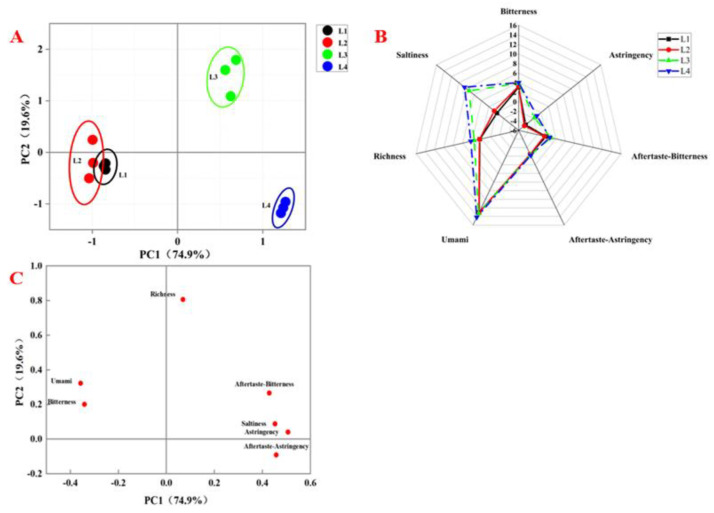
E-tongue analysis. (**A**) PCA plot of the samples at different processing stages. (**B**) Radar graph of the samples at different processing stages. (**C**) PCA loading plot of the samples at different processing stages.

**Table 1 foods-11-00963-t001:** Volatile compounds at different processing stages of Daokou braised chicken.

NO.	Compounds	Relative Percentage Content (%)			
		L1	L2	L3	L4
1	(Z)-3-Nonen-2-ol	-	0.02 ± 0.00 ^b^	0.07 ± 0.01 ^a^	0.06 ± 0.01 ^a^
2	1-Penten-3-ol	0.05 ± 0.01 ^b^	0.09 ± 0.00 ^b^	-	0.52 ± 0.58 ^a^
3	3-Methyl-1-butanol	0.34 ± 0.04 ^b^	0.28 ± 0.06 ^b^	1.89 ± 0.52 ^a^	-
4	1-Pentanol	1.84 ± 0.13	3.37 ± 0.79	2.78 ± 0.86	2.16 ± 0.47
5	1-Hexanol	8.59 ± 0.35 ^b^	7.88 ± 0.74 ^b^	11.57 ± 1.05 ^a^	7.92 ± 2.17 ^b^
6	1-Octen-3-ol	7.20 ± 0.05 ^b^	11.80 ± 0.34 ^a^	7.65 ± 1.32 ^b^	9.25 ± 0.01 ^a,b^
7	1-Heptanol	2.39 ± 0.23 ^a^	2.00 ± 0.08 ^a^	0.85 ± 0.02 ^b^	0.63 ± 0.27 ^b^
8	2,6-Dimethyl-4-heptanol	-	0.24 ± 0.01	0.77 ± 0.18	0.29 ± 0.25
9	2,4-Dimethyl-cyclohexanol	0.35 ± 0.14 ^b^	0.91 ± 0.12 ^a^	0.84 ± 0.06 ^a^	0.38 ± 0.06 ^b^
10	1-Octanol	3.82 ± 0.49 ^a^	2.88 ± 0.21 ^b^	1.07 ± 0.08 ^c^	0.81 ± 0.00 ^c^
11	(E)-2-Octen-1-ol	1.22 ± 0.04 ^b^	2.50 ± 0.47 ^a^	1.18 ± 0.05 ^b^	1.52 ± 0.13 ^b^
12	1-Nonen-4-ol	0.55 ± 0.00 ^c^	2.10 ± 0.49 ^a^	1.92 ± 0.31 ^a,b^	1.20 ± 0.06 ^b,c^
13	1-Nonanol	0.74 ± 0.17 ^b^	1.25 ± 0.27 ^a^	0.66 ± 0.06 ^b^	0.31 ± 0.02 ^b^
14	2-Ethyl-1-hexanol	0.17 ± 0.05	-	-	-
15	(E, Z)-2,4-Decadien-1-ol	0.06 ± 0.01 ^c^	0.21 ± 0.01 ^a^	-	0.15 ± 0.03 ^b^
Total alcohols		27.28 ± 0.23 ^b^	35.50 ± 3.34 ^a^	31.21 ± 4.34 ^a,b^	25.16 ± 1.96 ^b^
16	2-Butanone	-	-	-	0.14 ± 0.02
17	2,3-Butanedione	-	-	2.52 ± 0.10	2.86 ± 0.72
18	2,3-Pentanedione	-	0.06 ± 0.01 ^b^	0.29 ± 0.02 ^a^	0.30 ± 0.04 ^a^
19	2,3-Heptanedione	0.48 ± 0.52	0.10 ± 0.03	0.47 ± 0.11	0.81 ± 0.04
20	6-Methyl-2-Heptanone	0.07 ± 0.04	0.11 ± 0.06	-	-
21	3-Hydroxy-2-butanone	0.27 ± 0.08 ^b^	1.45 ± 0.21 ^b^	7.06 ± 0.43 ^a^	5.97 ± 1.51 ^a^
22	4-Octen-3-one	0.17 ± 0.02	0.13 ± 0.02	0.21 ± 0.03	0.25 ± 0.05
23	2,3-Octanedione	3.42 ± 0.78 ^c^	5.24 ± 0.49 ^b^	6.24 ± 0.02 ^b^	10.19 ± 0.18 ^a^
24	2-Nonanone	0.06 ± 0.01 ^b^	0.13 ± 0.01 ^b^	0.37 ± 0.11 ^a^	0.11 ± 0.04 ^b^
Total ketones		4.43 ± 0.27 ^c^	7.21 ± 0.83 ^c^	17.13 ± 0.39 ^b^	20.58 ± 1.98 ^a^
25	3-Methyl-butanal	-	-	-	0.20 ± 0.02
26	Pentanal	1.53 ± 0.02	1.31 ± 0.06	-	-
27	Hexanal	26.62 ± 0.16 ^a^	17.21 ± 1.10 ^b^	9.36 ± 2.78 ^c^	18.71 ± 2.76 ^b^
28	Heptanal	4.30 ± 0.52 ^a^	2.02 ± 0.33 ^b^	2.12 ± 0.23 ^b^	2.07 ± 0.08 ^b^
29	Octanal	6.12 ± 0.29 ^a^	2.39 ± 0.22 ^b^	0.86 ± 0.02 ^c^	1.19 ± 0.45 ^c^
30	Nonanal	15.18 ± 0.22 ^a^	6.31 ± 0.53 ^b^	3.86 ± 0.05 ^b^	5.12 ± 1.75 ^b^
31	2-Octenal	0.57 ± 0.14	0.79 ± 0.26	0.54 ± 0.09	0.56 ± 0.03
32	Decanal	0.70 ± 0.01	0.68 ± 0.13	-	-
33	Benzaldehyde	0.21 ± 0.04	0.19 ± 0.05	0.37 ± 0.04	1.18 ± 0.61
34	2-Nonenal	0.32 ± 0.07	0.32 ± 0.10	0.26 ± 0.06	0.34 ± 0.06
35	Z-4-Decenal	-	-	-	0.44 ± 0.05
36	Undecanal	0.06 ± 0.01 ^b^	0.14 ± 0.01 ^b^	0.67 ± 0.23 ^a^	0.32 ± 0.13 ^a,b^
37	(E, E)-2,4-Nonadienal	0.10 ± 0.01	0.11 ± 0.01	-	-
38	Trans-2-undecenal	0.27 ± 0.06	0.13 ± 0.02	-	-
39	(E, E)-2,4-decadienal	0.33 ± 0.00 ^b^	0.43 ± 0.03 ^a^	0.24 ± 0.03 ^c^	0.27 ± 0.02 ^c^
40	Tetradecanal	0.47 ± 0.07	-	0.58 ± 0.29	1.00 ± 0.35
Total aldehydes		56.76 ± 0.35 ^a^	32.02 ± 2.34 ^b^	18.82 ± 2.93 ^c^	31.51 ±6.13 ^b^
41	Decane	-	-	0.11 ± 0.01	0.08 ± 0.01
42	Methyl-benzene	0.23 ± 0.15 ^c^	0.60 ± 0.08 ^b^	1.05 ± 0.17 ^a^	1.02 ± 0.04 ^a^
43	Undecane	0.07 ± 0.01 ^b^	0.13 ± 0.04 ^a,b^	0.22 ± 0.04 ^a^	-
44	Ethylbenzene	0.06 ± 0.01 ^c^	0.12 ± 0.01 ^c^	0.36 ± 0.04 ^a^	0.24 ± 0.03 ^b^
45	1,3-Dimethyl-benzene	-	-	-	0.21 ± 0.01
46	(Z)-3-Dodecene	-	-	1.15 ± 0.57	-
47	Dodecane	-	0.12 ± 0.02 ^b^	0.42 ± 0.02 ^a^	0.32 ± 0.05 ^a^
48	1-Ethyl-4-methyl-benzene	-	-	-	0.20 ± 0.01
49	1-Ethyl-2-methyl-benzene	-	-	0.17 ± 0.04	-
50	1-Ethyl-3-methyl-benzene	-	-	-	0.11 ± 0.01
51	4,7-Methano-1H-indene, octahydro-	-	-	0.19 ± 0.01	0.21 ± 0.01
52	1,2,3-Trimethyl-benzene	-	-	0.12 ± 0.01	0.07 ± 0.01
53	Tridecane	-	0.56 ± 0.18	0.38 ± 0.25	0.44 ± 0.01
54	Tetradecane	-	0.66 ± 0.06	0.57 ± 0.04	0.53 ± 0.07
55	3-Ethyl-2-methyl-1,3-hexadiene	0.16 ± 0.07 ^b^	0.37 ± 0.04 ^a^	0.17 ± 0.01 ^b^	0.18 ± 0.02 ^b^
56	Nonadecane	-	-	1.25 ± 0.06	-
57	(2RS,3SR,4RS.5RS)-2,3: 4,5-Diepoxyhexan	-	2.13 ± 0.05	1.80 ± 0.22	1.60 ± 0.19
58	Heptadecane	0.04 ± 0.01 ^b^	0.07 ± 0.01 ^b^	0.39 ± 0.13 ^a,b^	0.23 ± 0.06 ^a^
59	4-Decene, 2,2-dimethyl-, (E)-	-	-	-	0.26 ± 0.10
60	Octadecane	-	-	0.16 ± 0.06	0.10 ± 0.01
Total hydrocarbons		0.55 ± 0.09 ^c^	4.73 ± 0.25 ^b^	8.47 ± 1.10 ^a^	5.62 ± 0.12 ^b^
61	Limonene	-	-	0.73 ± 0.01	0.72 ± 0.11
62	1,8-Cineole	-	1.05 ± 0.11 ^c^	11.28 ± 0.45 ^a^	4.38 ± 1.49 ^b^
63	Trans-p-mentha-1(7),8-dien-2-ol	-	-	-	0.06 ± 0.01
64	Linalool	-	-	-	0.23 ± 0.06
65	3-Cyclohexen-1-ol, 4-methyl-1-(1-methylethyl)-	-	-	0.13 ± 0.01	0.21 ± 0.01
66	α-Terpineol	-	-	0.08 ± 0.00	0.10 ± 0.01
67	β-Fenchol	-	-	0.80 ± 0.11	0.60 ± 0.15
68	Anethole	-	-	0.23 ± 0.04	0.14 ± 0.02
69	Trans-geraniol	-	-	0.42 ± 0.02	0.25 ± 0.01
70	(E)-Cinnamaldehyde	-	-	0.43 ± 0.01	0.28 ± 0.01
Total terpenes		-	1.05 ± 0.18 ^c^	14.07 ± 0.07 ^a^	6.93 ± 0.31 ^b^
71	Acetic acid, hexyl ester	-	0.27 ± 0.07	0.28 ± 0.04	0.20 ± 0.06
Total esters		-	0.27 ± 0.07	0.28 ± 0.04	0.20 ± 0.06
72	2-Pentyl-furan	0.29 ± 0.08 ^d^	1.09 ± 0.12 ^c^	1.38 ± 0.07 ^b^	1.98 ± 0.18 ^a^
73	2-Furanmethanol, 5-ethenyltetrahydro	-	0.27 ± 0.07 ^b^	1.32 ± 0.21 ^a^	2.10 ± 0.47 ^a^
74	Oxime-,methoxy-phenyl-	0.66 ± 0.49 ^a^	0.14 ± 0.01 ^b^	0.21 ± 0.07 ^b^	0.19 ± 0.03 ^b^
75	N,N-Dibutylformamide	-	0.07 ± 0.01	-	-
Total heterocycles		1.20 ± 0.25 ^d^	1.54 ± 0.26 ^c^	2.94 ± 0.52 ^b^	4.22 ± 0.45 ^a^

^1 a–d^: Means within the same row with different letters differ significantly (*p* < 0.05); ^2^ RI: the retention index calculated using n-alkanes C_6_–C_26_ as external standard; ^3^ MS: identification by comparison with mass spectra.

**Table 2 foods-11-00963-t002:** The names of compound in Figure 5 in regions “a, b, c and d”.

a	b	c	d
Ethyl-3-methyl butanoate	2-Methylpropionic acid	1-Butanol	Hydroxyacetone
Methyl-2-methyl butanoate	Ethanol	3-Methyl-3-butennol	Ethyl-3-methylbutanoate
Ethyl butanoate	Hydroxyacetone monomer	N-nitrosodimethylamine	Phenylacetaldehyde
Ethyl pentanoate	Pentan-2 3-dione	2-Hexenol	2-Pentyl-furan
Z-3-hexenol	2-Pentanone	2-Methyl pyrazine	Ethyl propionate
Butanoic acid	Isoamyl alcohol	Propyl acetate	Alpha-pinene
Pentanoic acid	2-Methylpyrazine	Benzaldehyde	2-Methoxyphenol
	Hexanal	Pentanol	Acetophenone
	P-xylene	Heptanal	Isovaleric acid
	Hexanol		2-Octanone
	2-Furanmethanol-furfuryl alcohol		2 3-Butanediol
	Ethyl pentanoate		2-Hexanone
			3-Methylpentanoic acid
			nonanal
			2-Phenylethanol
			Phenol-4-methyl
			1-Octanol
			1 2-Furanyl-ethanone
			2-Nonanone
			Alpha-pinene

Note: The numbers represent the names of compounds that could not be accurately identified. The names of compounds are listed from left to right in regions a, b, c and d.

## Data Availability

The data presented in this study are available on request from the corresponding author.

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
