# Peer review of "Multiple Technologies Combined to Analyze the Changes of Odor and Taste in Daokou Braised Chicken during Processing"

_foods, 2022, doi:10.3390/foods11070963_

Round 1

Reviewer 1 Report

ENTRO il 24 FEBBRAIO 2022

The authors examine the changes of volatile compounds in Daokou braised chicken; raw samples are also studied. GC-MS, GC-IMS, e-nose and e-tongue are used to evaluate the flavor profiles in the samples.

The combination of several analytical techniques enables a more complete picture of the volatile compounds in chicken to be obtained in Daokou braised chicken during the cooking process.

However, some observations are necessary. I am aware that the design of the manuscript is based on a complex analytical assessment of the Daokou braised chicken.

Furthermore, it can be observed the absence of some considerations concerning the origin of certain volatile compounds in the different samples.

The authors did not specify whether the combination of techniques has a positive impact from a practical point of view. It would also have been appropriate to briefly consider the cost-benefit ratio to be applied to the analytical evaluation of Daokou braised chicken. This product is “famous for distinctive flavor, beautiful shape, bright color, tender and boneless” and “deeply loved by consumers”, as we can see in the manuscript.

Some observation will follow:

some English sentences can be improved (e.g. line 92)

Abstract: the aim of the study needs to be reported

Line 47: explain the acronyms of IMP, Glu, Lys

Line 86: move section 2.2. to a paragraph

Line 93: Tangerine can be written using lowercase; “angelica daurica” must be written as a plant species, and in italics (Angelica daurica)

98: did the breasts and thighs samples include skin? Please, it would be better to specify this detail

Figure 1: it is not clearly explained; for example, pictures of spices would need a separate caption.

Furthermore, in my opinion, the caption of figure 1 should be completed with a reference to the images on the right-hand side of the page.

Overall, the images included in Figure 1 are very close to each other

Line 105: why the chilled samples were not equilibrated to room temperature before extraction?

Line 106: it would be better to specify the fiber’s producer,

Lines 189-190: the Ketones “2,3-octanedione and 3-hydroxy- 2-butanone” (Ketons) in Table 1 are included between the total alcohols

Line 196:” 3-methyl-butanal may production”; the sentence must be changed

Lines 189-190-193-194- 201 and so on: it would be better if the compounds’ name were uniformly written in the text and tables (i.e. 3-methyl-butanal; 2,6-Dimethyl-4-heptanol, etc)

From line 218 to line 240: the authors report the results related to terpenes, heterocyclic compounds and esters together with some considerations on their origins (i.e. terpenes). On the contrary, these considerations were not reported for other compounds.

It is advisable to review the content of the paragraph, by adding further considerations. Consequently, the paragraph name can be “Results and Discussion”. In fact the manuscript has no “Discussion”.

Line 243: Figure 2- the colours that distinguish compounds in the graph on the left-hand side of the page do not match the colours put in the graph on the right side

Figure 5: it is very difficult to distinguish the words and numbers in “the fingerprint spectra (C) of volatile compounds”

Line 418:Conclusions - most sentences refer more to a discussion than to the conclusions section.

Some changes in the layout of the results are recommended; a section for Discussion would be necessary. In this way conclusions could be more concise.

Author Response

The  replies to reviewers has been submitted as attachments.

Reviewer 2 Report

This paper will contribute to the knowledge on flavour evolution during the processing of Daokuo braised chicken by using intelligent sensor techniques as electronic nose and electronic tongue combined with GC-MS and GC-IMS chromatography analyses. Even if there are other works already published, this food product is not for international use since it is highly specific for Chinese traditions. The aim of the study, as also specified in the title, regards the combination of multi-technologies to analyse the change of odour and taste in Daokou braised chicken during processing, but the authors discuss on them separately, there are no combination of analytical and sensor response data and so no correlation between them.

The work is not well written and needed to be revised by an English mother tongue.

Therefore, I suggest the manuscript major revision:

ABSTRACT

Line 9: “Volatile odor”. Volatile compounds or odour?

MATERIALS AND METHODS

Line 86-87: Delete the subtitle.

Line 92: Which kind of oil was used?

Line 93: How were the 8 spices used?

Figure 1: This figure is a graphical abstract; it is not a key technology and processing.

Line 105: Specify the sample treatment.

2.5 E-nose analysis: describe the electronic nose sensors.

Line 140: r/min? What the authors mean?

Line 143-144: Replace aftertaste –A and aftertaste –B with aftertaste –Astringency and aftertaste –bitterness.

RESULTS AND DISCUSSION

Insert the table and the figure when cited in the text.

Discuss the figure 2 after the table 1, in order to make the discussion of the results more fluent and easier to read.

Concerning the electronic nose and electronic tongue PCA results, it is interesting to know something on how the sensors influence and/or contribute to the discrimination of the samples. This can be shown by PCA loading plot. It is also interesting to combine statistically the analytical, e-nose and e-tongue data to visualize the correlation and to provide a flavour profile.

Line 406: “Richness” what this sensor mean?

Figure 8: how is it possible that there is no difference between raw and fried chicken? how do the authors explain it?

CONCLUSIONS:

Line 439-440: “The combination of GC-MS, GC-IMS, e-nose and e-tongue could provide more comprehensive flavour profile in Daokou braised chicken during processing.” There are no combination of analytical and sensor response data so they do not provide a profile of Daokou braised chicken during processing.

Author Response

(The authors gave the same response as above.)

Reviewer 3 Report

The manuscript entitled „Multi-technologies combined to analyze the changes of odor and 2 taste in Daokou braised chicken during processing Distribution and quantification of 1,2-Propylene glycol enantiomers 2 in Baijiu” is of great interest and scientific value.

However, I have a few comments and questions.

  • Why was raw chicken subjected to sensory analyses (profile of odour compounds)? It is obvious that during the thermal treatment of meat, reactions take place which lead to the formation of compounds that confer sensory qualities to the final product. Therefore, the profile of the aroma compounds in the raw product and after heat treatment will be different.
  • On what basis was the fibre selected for SPME?
  • Supplementary material (chromatograms) would enrich the article.
  • What is the practical significance of the research carried out?

Overall, the manuscript needs a minor revision.

Author Response

(The authors gave the same response as above.)

Round 2

Reviewer 2 Report

Thanks to authors for the reviewing of the manuscript entitled “Multi-technologies combined to analyze the changes of odor and taste in Daokou braised chicken during processing” (ID: foods-1613108). All the comments and suggested corrections were made in order to improve the manuscript. Therefore, I suggest the publication of the manuscript on Foods.